# MIRROR-GENERATIVE NEURAL MACHINE TRANSLATION

**Zaixiang Zheng**[1]**, Hao Zhou**[2]**, Shujian Huang**[1]**, Lei Li**[2]**, Xin-Yu Dai**[1]**, Jiajun Chen**[1]
[1]National Key Laboratory for Novel Software Technology, Nanjing University
`zhengzx@smail.nju.edu.cn`,{`huangsj,daixinyu,chenjj`}`@nju.edu.cn`
[2]ByteDance AI Lab
{`zhouhao.nlp,lileilab`}`@bytedance.com`

## ABSTRACT

Training neural machine translation models (NMT) requires a large amount of parallel corpus, which is scarce for many language pairs. However, raw non-parallel corpora are often easy to obtain. Existing approaches have not exploited the full potential of non-parallel bilingual data either in training or decoding. In this paper, we propose the mirror-generative NMT (MGNMT), a single unified architecture that simultaneously integrates the source to target translation model, the target to source translation model, and two language models. Both translation models and language models share the same latent semantic space, therefore both translation directions can learn from non-parallel data more effectively. Besides, the translation models and language models can collaborate together during decoding. Our experiments show that the proposed MGNMT consistently outperforms existing approaches in a variety of language pairs and scenarios, including resource-rich and low-resource situations.

## 1 INTRODUCTION

Neural machine translation (NMT) systems (Sutskever et al., 2014; Bahdanau et al., 2015; Gehring et al., 2017; Vaswani et al., 2017) have given quite promising translation results when abundant parallel bilingual data are available for training. But obtaining such large amounts of parallel data is non-trivial in most machine translation scenarios. For example, there are many low-resource language pairs (e.g., English-to-Tamil), which lack adequate parallel data for training. Moreover, it is often difficult to adapt NMT models to other domains if there is only limited test domain parallel data (e.g., medical domain), due to the large domain discrepancy between the test domain and the parallel data for training (usually news-wires). For these cases where the parallel bilingual data are not adequate, making the most use of non-parallel bilingual data (always quite cheap to get) is crucial to achieving satisfactory translation performance.

We argue that current NMT approaches of exploiting non-parallel data are not necessarily the best, in both training and decoding phases. For training, back-translation (Sennrich et al., 2016b) is the most widely used approach for exploiting monolingual data. However, back-translation individually updates the two directions of machine translation models, which is not the most effective. Specifically, given monolingual data $x$ (of source language) and $y$ (of target language)[1], back-translation utilizes $y$ by applying *tgt2src* translation model ($\text{TM}_{y \to x}$) to get predicted translations $\hat{x}$. Then the pseudo translation pairs $\langle \hat{x}, y \rangle$ are used to update the *src2tgt* translation model ($\text{TM}_{x \to y}$). $x$ can be used in the same way to update $\text{TM}_{y \to x}$. Note that here $\text{TM}_{y \to x}$ and $\text{TM}_{x \to y}$ are independent and updated individually. Namely, each updating of $\text{TM}_{y \to x}$ will not directly benefit $\text{TM}_{x \to y}$. Some related work like *joint back-translation* Zhang et al. (2018) and *dual learning* He et al. (2016a) introduce iterative training to make $\text{TM}_{y \to x}$ and $\text{TM}_{x \to y}$ benefit from each other implicitly and iteratively. But translation models in these approaches are still independent. Ideally, gains from non-parallel data can be enlarged if we have relevant $\text{TM}_{y \to x}$ and $\text{TM}_{x \to y}$. In that case, after every updating of $\text{TM}_{y \to x}$, we may directly get better $\text{TM}_{x \to y}$ and vice versa, which exploits non-parallel data more effectively.

---

[1]Please refer to Section 2 for the notation in details.

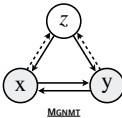

Figure 1: The graphical model of MGNMT.

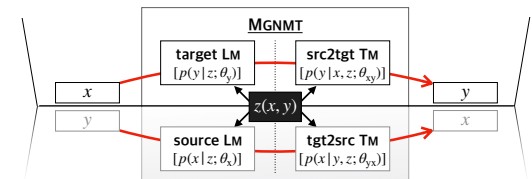

Figure 2: Illustration of the *mirror* property of MGNMT.

For decoding, some related works (Gulcehre et al., 2015) propose to interpolate external language models $LM_y$ (trained separately on target monolingual data) to translation model $TM_{x \to y}$, which includes knowledge from target monolingual data for better translation. This is particularly useful for domain adaptation because we may obtain better translation output quite fitting test domain (e.g., social networks), through a better $LM_y$. However, directly interpolating an independent language model in decoding maybe not the best. First, the language model used here is external, still independently learned to the translation model, thus the two models may not cooperate well by a simple interpolation mechanism (even conflict). Additionally, the language model is only included in decoding, which is not considered in training. This leads to the inconsistency of training and decoding, which may harm the performance.

In this paper, we propose the mirror-generative NMT (MGNMT) to address the aforementioned problems for effectively exploiting non-parallel data in NMT. MGNMT is proposed to jointly train translation models (i.e., $TM_{x \to y}$ and $TM_{y \to x}$) and language models (i.e., $LM_x$ and $LM_y$) in a unified framework, which is non-trivial. Inspired by generative NMT (Shah & Barber, 2018), we propose to introduce a latent semantic variable $z$ shared between $x$ and $y$. Our method exploits the symmetry, or *mirror* property, in decomposing the conditional joint probability $p(x, y|z)$, namely:

$$\log p(x, y|z) = \log p(x|z) + \log p(y|x, z) = \log p(y|z) + \log p(x|y, z)$$

$$= \frac{1}{2} [\underbrace{\log p(y|x, z)}_{src2tgt\ TM_{x \to y}} + \underbrace{\log p(y|z)}_{target\ LM_y} + \underbrace{\log p(x|y, z)}_{tgt2src\ TM_{y \to x}} + \underbrace{\log p(x|z)}_{source\ LM_x}] \quad (1)$$

The graphical model of MGNMT is illustrated in Figure 1. MGNMT aligns the bidirectional translation models as well as language models in two languages through a shared latent semantic space (Figure 2), so that all of them are relevant and become conditional independent given $z$. In such case, MGNMT enables following advantages:

(i) For training, thanks to $z$ as a bridge, $TM_{y \to x}$ and $TM_{x \to y}$ are not independent, thus every updating of one direction will directly benefit the other direction. This improves the efficiency of using non-parallel data. (Section 3.1)

(ii) For decoding, MGNMT could naturally take advantages of its internal target-side language model, which is jointly learned with the translation model. Both of them contribute to the better generation process together. (Section 3.2)

Note that MGNMT is orthogonal to dual learning (He et al., 2016a) and joint back-translation (Zhang et al., 2018). Translation models in MGNMT are dependent, and the two translation models could directly promote each other. Differently, dual learning and joint back-translation works in an implicit way, and these two approaches can also be used to further improve MGNMT. The language models used in dual learning faces the same problem as Gulcehre et al. (2015). Given GNMT, the proposed MGNMT is also non-trivial. GNMT only has a source-side language model, thus it cannot enhance decoding like MGNMT. Also, in Shah & Barber (2018), they require GNMT to share all the parameters and vocabularies between translation models so as to utilize monolingual data, which is not best suited for distant language pairs. We will give more comparison in the related work.

Experiments show that MGNMT achieves competitive performance on parallel bilingual data, while it does advance training on non-parallel data. MGNMT outperforms several strong baselines in different scenarios and language pairs, including resource-rich scenarios, as well as resource-poor circumstances on low-resource language translation and cross-domain translation. Moreover, we show that translation quality indeed becomes better when the jointly learned translation model and language model of MGNMT work together. We also demonstrate that MGNMT is architecture-free which can be applied to any neural sequence model such as Transformer and RNN. These pieces of evidence verify that MGNMT meets our expectation of fully utilizing non-parallel data.

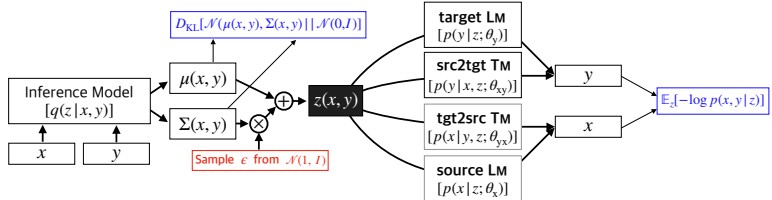

Figure 3: Illustration of the architecture of MGNMT.

## 2 BACKGROUND AND RELATED WORK

**Notation**   Given a pair of sentences from source and target languages, e.g., $\langle x, y \rangle$, we denote $x$ as a sentence of the "*source*" language, and $y$ as a sentence of the "*target*" language. Additionally, we use the terms "*source-side*" and "*target-side*" of a translation direction to denote the input and the output sides of it, e.g., the source-side of the "*tgt2src*" translation is the *target* language.

**Neural machine translation**   Conventional neural machine translation (NMT) models often adopt an *encoder-decoder* framework (Bahdanau et al., 2015) with discriminative learning. Here NMT models aim to approximate the conditional distribution $\log p(y|x; \theta_{xy})$ over a target sentence $y = \langle y_1, \ldots, y_{L_y} \rangle$ given a source sentence $x = \langle x_1, \ldots, x_{L_x} \rangle$. Here we refer to such regular NMT models as discriminative NMT models. Training criterion for a discriminative NMT model is to maximize the conditional log-likelihood $\log p(y|x; \theta_{xy})$ on abundant parallel bilingual data $\mathcal{D}_{xy} = \{x^{(n)}, y^{(n)}|n = 1...N\}$ of i.i.d observations.

As pointed out by Zhang et al. (2016) and Su et al. (2018), the shared semantics $z$ between $x$ and $y$ are learned in an implicit way in discriminative NMT, which is insufficient to model the *semantic equivalence* in translation. Recently, Shah & Barber (2018) propose a generative NMT (GNMT) by modeling the joint distribution $p(x, y)$ instead of $p(y|x)$ with a latent variable $z$:

$$\log p(x, y|z; \boldsymbol{\theta} = \{\theta_x, \theta_{xy}\}) = \log p(x|z; \theta_x) + \log p(y|x, z; \theta_{xy})$$

where GNMT models $\log p(x|z; \theta_x)$ as a source variational language model. Eikema & Aziz (2019) also propose a similar approach. In addition, Chan et al. (2019) propose a generative insertion-based modeling for sequence, which also models the joint distribution.

**Exploiting non-parallel data for NMT**   Both discriminative and generative NMT could not directly learn from non-parallel bilingual data. To remedy this, back-translation and its variants (Sennrich et al., 2016b; Zhang et al., 2018) exploit non-parallel bilingual data by generating synthetic parallel data. Dual learning (He et al., 2016a; Xia et al., 2017) learns from non-parallel data in a round-trip game via reinforcement learning, with the help of pretrained language models. Although these methods have shown their effectiveness, the independence between translation models, and between translation and language models (dual learning) may lead to inefficiency to utilize non-parallel data for both training and decoding as MGNMT does. In the meantime, iterative learning schemes like them could also complement MGNMT.

Some other related studies exploit non-parallel bilingual data by sharing all parameters and vocabularies between source and target languages, by which two translation directions can be updated by either monolingual data (Dong et al., 2015; Johnson et al., 2017; Firat et al., 2016; Artetxe et al., 2018; Lample et al., 2018a;b), and GNMT as well in an auto-encoder fashion. However, they may still fail to apply to distant language pairs (Zhang & Komachi, 2019) such as English-to-Chinese due to the potential issues of non-overlapping alphabets, which is also verified in our experiments.

Additionally, as aforementioned, integrating language model is another direction to exploit monolingual data (Gulcehre et al., 2015; Stahlberg et al., 2018; Chu & Wang, 2018) for NMT. However, this kind of methods often resorts to external trained language models, which is agnostic to translation task. Besides, although GNMT contains a source-side language model, it cannot help decoding. In contrast, MGNMT jointly learns translation and language modeling probabilistically and can naturally rely on both together for a better generation.

---

**Algorithm 1** Training MGNMT from Non-Parallel Data

---

**Input:** (pretrained) MGNMT $\mathcal{M}(\boldsymbol{\theta})$ , source monolingual dataset $\mathcal{D}_\mathrm{x}$, target monolingual dataset $\mathcal{D}_\mathrm{y}$

1: **while** not converge **do**
2:     Draw source and target sentences from non-parallel data: $x^{(s)} \sim \mathcal{D}_\mathrm{x}$, $y^{(t)} \sim \mathcal{D}_\mathrm{y}$
3:     Use $\mathcal{M}$ to translate $x^{(s)}$ to construct a pseudo-parallel sentence pair $\langle x^{(s)}, y_\mathrm{pseu}^{(s)}\rangle$
4:     Compute $\mathcal{L}(x^{(s)}; \theta_\mathrm{x}, \theta_\mathrm{yx}, \phi)$ with $\langle x^{(s)}, y_\mathrm{pseu}^{(s)}\rangle$ by Equation (5)
5:     Use $\mathcal{M}$ to translate $y^{(t)}$ to construct a pseudo-parallel sentence pair $\langle x_\mathrm{pseu}^{(t)}, y^{(t)}\rangle$
6:     Compute $\mathcal{L}(y^{(t)}; \theta_\mathrm{y}, \theta_\mathrm{xy}, \phi)$ with $\langle x_\mathrm{pseu}^{(t)}, y^{(t)}\rangle$ by Equation (4)
7:     Compute the deviation $\nabla\boldsymbol{\theta}$ by Equation (6)
8:     Update parameters $\boldsymbol{\theta} \rightarrow \boldsymbol{\theta} + \eta\nabla\boldsymbol{\theta}$
9: **end while**

---

# 3 MIRROR-GENERATIVE NEURAL MACHINE TRANSLATION

We propose the mirror-generative NMT (MGNMT), a novel deep generative model which simultaneously models a pair of *src2tgt* and *tgt2src* (variational) translation models, as well as a pair of source and target (variational) language models, in a highly integrated way with the mirror property. As a result, MGNMT can learn from non-parallel bilingual data, and naturally interpolate its learned language model with the translation model in the decoding process.

The overall architecture of MGNMT is illustrated graphically in Figure 3. MGNMT models the joint distribution over the bilingual sentences pair by exploiting the *mirror* property of the joint probability: $\log p(x,y|z) = \frac{1}{2}[\log p(y|x,z) + \log p(y|z) + \log p(x|y,z) + \log p(x|z)]$, where the latent variable $z$ (we use a standard Gaussian prior $z \sim \mathcal{N}(0,I)$) stands for the shared semantics between $x$ and $y$, serving as a bridge between all the integrated translation and language models.

## 3.1 TRAINING

### 3.1.1 LEARNING FROM PARALLEL DATA

We first introduce how to train MGNMT on a regular parallel bilingual data. Given a parallel bilingual sentence pair $\langle x, y\rangle$, we use stochastic gradient variational Bayes (SGVB) (Kingma & Welling, 2014) to perform approximate maximum likelihood estimation of $\log p(x,y)$. We parameterize the approximate posterior $q(z|x,y;\phi) = \mathcal{N}(\boldsymbol{\mu}_\phi(x,y), \boldsymbol{\Sigma}_\phi(x,y))$. Then from Equation (1), we can have the Evidence Lower BOund (ELBO) $\mathcal{L}(x,y;\boldsymbol{\theta};\phi)$ of the log-likelihood of the joint probability as:

$$\log p(x,y) \geq \mathcal{L}(x,y;\boldsymbol{\theta},\phi) = \mathbb{E}_{q(z|x,y;\phi)}[\frac{1}{2}\{\log p(y|x,z;\theta_\mathrm{xy}) + \log p(y|z;\theta_\mathrm{y})$$
$$+ \log p(x|y,z;\theta_\mathrm{yx}) + \log p(x|z;\theta_\mathrm{x})\}] \qquad (2)$$
$$- D_\mathrm{KL}[q(z|x,y;\phi)||p(z)]$$

where $\boldsymbol{\theta} = \{\theta_\mathrm{x}, \theta_\mathrm{yx}, \theta_\mathrm{y}, \theta_\mathrm{xy}\}$ is the set of the parameters of translation and language models. The first term is the (expected) log-likelihood of the sentence pair. The expectation is obtained by Monte Carlo sampling. The second term is the KL-divergence between $z$'s approximate posterior and prior distributions. By relying on a reparameterization trick (Kingma & Welling, 2014), we can now jointly train all the components using gradient-based algorithms.

### 3.1.2 LEARNING FROM NON-PARALLEL DATA

Since MGNMT has intrinsically a pair of mirror translation models, we design an *iterative* training approach to exploit non-parallel data, in which both directions of MGNMT could benefit from the monolingual data mutually and boost each other. The proposed training process on non-parallel bilingual data is illustrated in Algorithm 1.

Formally, given non-parallel bilingual sentences, i.e., $x^{(s)}$ from source monolingual dataset $\mathcal{D}_\mathrm{x} = \{x^{(s)}|s = 1...S\}$ and $y^{(t)}$ from target monolingual dataset $\mathcal{D}_\mathrm{y} = \{y^{(t)}|t = 1...T\}$, we aim to maximize the lower-bounds of the likelihood of their marginal distributions mutually:

$$\log p(x^{(s)}) + \log p(y^{(t)}) \geq \mathcal{L}(x^{(s)}; \theta_\mathrm{x}, \theta_\mathrm{yx}, \phi) + \mathcal{L}(y^{(t)}; \theta_\mathrm{y}, \theta_\mathrm{xy}, \phi) \qquad (3)$$

where $\mathcal{L}(x^{(s)}; \theta_x, \theta_{yx}, \phi)$ and $\mathcal{L}(y^{(t)}; \theta_y, \theta_{xy}, \phi)$ are the lower-bounds of the source and target marginal log-likelihoods, respectively.

Let us take $\mathcal{L}(y^{(t)}; \theta_y, \theta_{xy}, \phi)$ for example. Inspired by Zhang et al. (2018), we sample $x$ with $p(x|y^{(t)})$ in source language as $y^{(t)}$'s translation (i.e., back-translation) and obtain a pseudo-parallel sentence pair $\langle x, y^{(t)} \rangle$. Accordingly, we give the form of $\mathcal{L}(y^{(t)}; \theta_y, \theta_{xy}, \phi)$ in Equation (4). Likewise, Equation (5) is for $\mathcal{L}(y^{(t)}; \theta_y, \theta_{xy}, \phi)$. (See Appendix for the their derivation).

$$
\begin{aligned}
\mathcal{L}(y^{(t)}; \theta_y, \theta_{xy}, \phi) = \mathbb{E}_{p(x|y^{(t)})} \big[ \mathbb{E}_{q(z|x, y^{(t)}; \phi)} [ \frac{1}{2} \{ \log p(y^{(t)}|z; \theta_y) + \log p(y^{(t)}|x, z; \theta_{xy}) \}] \\
- D_{\mathrm{KL}}[q(z|x, y^{(t)}; \phi) || p(z)]]
\end{aligned}
\tag{4}
$$

$$
\begin{aligned}
\mathcal{L}(x^{(s)}; \theta_x, \theta_{yx}, \phi) = \mathbb{E}_{p(y|x^{(s)})} \big[ \mathbb{E}_{q(z|x^{(s)}, y; \phi)} [ \frac{1}{2} \{ \log p(x^{(s)}|z; \theta_x) + \log p(x^{(s)}|y, z; \theta_{yx}) \}] \\
- D_{\mathrm{KL}}[q(z|x^{(s)}, y; \phi) || p(z)]]
\end{aligned}
\tag{5}
$$

The parameters included in Equation (3) can be updated via gradient-based algorithm, where the deviations are computed as Equation (6) in a mirror and integrated behavior:

$$
\nabla \boldsymbol{\theta} = \nabla_{\{\theta_x, \theta_{yx}\}} \mathcal{L}(x^{(s)}; \cdot) + \nabla_{\{\theta_y, \theta_{xy}\}} \mathcal{L}(y^{(t)}; \cdot) + \nabla_\phi [\mathcal{L}(x^{(s)}; \cdot) + \mathcal{L}(y^{(t)}; \cdot)]
\tag{6}
$$

The overall training process of exploiting non-parallel data does to some extent share a similar idea with joint back-translation (Zhang et al., 2018). But they only utilize one side of non-parallel data to update one direction of translation models for each iteration. Thanks to $z$ from the shared approximate posterior $q(z|x, y; \phi)$ as a bridge, both directions of MGNMT could benefit from either of the monolingual data. Besides, MGNMT's "back-translated" pseudo translations have been improved by advanced decoding process (see Equation (7)), which leads to a better learning effect.

## 3.2 DECODING

Thanks to simultaneously modeling of translation models and language models, MGNMT is now able to generate translation by the collaboration of translation and language models together. This endows MGNMT's translation in target-side language with more domain-related fluency and quality.

Due to the mirror nature of MGNMT, the decoding process is also of symmetry: given a source sentence $x$ (or target sentence $y$), we want to find a translation by $y = \mathrm{argmax}_y \, p(y|x) = \mathrm{argmax}_y \, p(x, y)$ ($x = \mathrm{argmax}_x \, p(x|y) = \mathrm{argmax}_x \, p(x, y)$), which is approximated by a mirror variant of the idea of EM decoding algorithm in GNMT (Shah & Barber, 2018). Our decoding process is illustrated in Algorithm 2.

Let's take the *srg2tgt* translation as example. Given a source sentence $x$, 1) we first samples an initial $z$ from the standard Gaussian prior and then obtain an initial *draft translation* as $\tilde{y} = \mathrm{argmax}_y \, p(y|x, z)$; 2) this translation is iteratively refined by re-sampling $z$ this time from the approximate posterior $q(z|x, \tilde{y}; \phi)$, and re-decoding with beam search by maximizing the ELBO:

$$
\begin{aligned}
\tilde{y} \leftarrow & \mathrm{argmax}_y \, \mathcal{L}(x, \tilde{y}; \boldsymbol{\theta}, \phi) \\
& = \mathrm{argmax}_y \, \mathbb{E}_{q(z|x, \tilde{y}; \phi)} [\log p(y|x, z) + \log p(y|z) + \log p(x|z) + \log p(x|y, z)] \\
& = \mathrm{argmax}_y \, \mathbb{E}_{q(z|x, \tilde{y}; \phi)} \big[ \sum_i [\underbrace{\log p(y_i|y_{<i}, x, z) + \log p(y_i|y_{<i}, z)}_{\text{Decoding Score}}] + \underbrace{\log p(x|z) + \log p(x|y, z)}_{\text{Reconstructive Reranking Score}} \big]
\end{aligned}
\tag{7}
$$

The *decoding scores* at each step are now given by $\mathrm{TM}_{x \to y}$ and $\mathrm{LM}_y$, which is helpful to find a sentence $y$ not only being the translation of $x$ but also being more possible in the target language[2]. The *reconstructive reranking scores* are given by $\mathrm{LM}_x$ and $\mathrm{TM}_{y \to x}$, which are employed after translation candidates are generated. MGNMT can leverage this kind of scores to sort the translation candidates and determine the most faithful translation to the source sentence. It is to essentially share the

---

[2]Empirically, we find that using $\log p(y_i|y_{<i}, x, z) + \beta \log p(y_i|y_{<i}, z)$ with a coefficient $\beta \approx 0.3$ leads to more robust results, which shares the similar observations with Gulcehre et al. (2015).

---

**Algorithm 2** MGNMT Decoding with EM Algorithm

---

**Input:** MGNMT $\mathcal{M}(\boldsymbol{\theta})$, input sentence $x$, input language $l$
**Output:** $x$'s translation $y$
**procedure:** DECODING$(x, l)$
1: **if** $l$ is the "`target`" language **then**
2:     Swap the parameters of $\mathcal{M}(\boldsymbol{\theta})$ regarding language: $\{\theta_x, \theta_{yx}\} \leftrightarrow \{\theta_y, \theta_{xy}\}$
3: **end if**
4: $y = \text{RUN}(x)$
5: **return** translation $y$

**procedure:** RUN$(x)$
1: Sample $z$ from standard Gaussian: $z \sim \mathcal{N}(0, I)$
2: Generate initial draft translation: $\tilde{y} = \text{argmax}_y \ \log p(y|x, z)$
3: **while** not converage **do**
4:     Sample $\mathbf{z} = \{z^{(k)}\}_{s=1}^K$ from variational distribution: $z^{(k)} \sim q(z|x, \tilde{y})$
5:     Generate translation candidates $\{\hat{y}\}$ via beam search by maximizing $\frac{1}{K} \sum_{z^{(k)}} [\sum_i \log p(y_i|y_{<i}, x, z^{(k)}) + \log p(y_i|y_{<i}, z^{(k)})]$ ▷ *"decoding scores" in Equation (7)*
6:     Determine the best intermediate translation $\tilde{y}$ via ranking $\{\hat{y}\}$ by maximizing $\frac{1}{K} \sum_{z^{(k)}} [\log p(x|z^{(k)}) + \log p(x|y, z^{(k)})]$ ▷ *"reconstructive reranking scores" in Equation (7)*
7: **end while**
8: **return** translation $y = \tilde{y}$

---

Table 1: Statistics of datasets for each translation tasks.

| Dataset | WMT14 EN↔DE | NIST EN↔ZH | WMT16 EN↔RO | IWSLT16 EN↔DE |
|---|---|---|---|---|
| Parallel | 4.50m | 1.34m | 0.62m | 0.20m (TED) |
| Non-parallel | 5.00m | 1.00m | 1.00m | 0.20m (NEWS) |
| Dev/Test | newstest2013/14 | MT06/MT03 | newstest2015/16 | tst13/14&newstest2014 |

same idea as Ng et al. (2019) and Chen et al. (2019), which propose a neural noisy channel reranking to incorporate reconstructive score to rerank the translation candidates. Some studies like Tu et al. (2017), Cheng et al. (2016) also exploit this bilingual semantic equivalence as reconstruction regularization for training.

# 4 EXPERIMENT

**Dataset** To evaluate our model in resource-poor scenarios, we conducted experiments on WMT16 English-to/from-Romanian (WMT16 EN↔RO) translation task as low-resource translation and IWSLT16 English-to/from-German (IWSLT16 EN↔DE) parallel data of TED talk as cross-domain translation. As for resource-rich scenarios, we conducted experiments on WMT14 English-to/from-German (WMT14 EN↔DE), NIST English-to/from-Chinese (NIST EN↔ZH) translation tasks. For all the languages, we use the non-parallel data from News Crawl, except for NIST EN↔ZH, where the Chinese monolingual data were extracted from LDC corpus. Table 1 lists the statistics. In particular, for cross-domain translation, all models are trained using parallel data from TED domain, and non-parallel data from NEWS domain if applicable. We then evaluate these models on both TED and NEWS testsets, respectively.

**Experimental settings** We implemented our models on the top of Transformer (Vaswani et al., 2017) and RNMT (Bahdanau et al., 2015) and GNMT (Shah & Barber, 2018) as well on Pytorch[3]. In this section, we only compare experimental results on Transformer implementation.[4]

For all languages pairs, sentence were encoded using byte pair encoding (Sennrich et al., 2016a, BPE) with 32k merge operations, jointly learned from the concatenation of the *parallel* training dataset only (except for NIST ZH-EN whose BPEs were learned separately). We used the Adam optimizer (Kingma & Ba, 2014) with the same learning rate schedule strategy as Vaswani et al. (2017) with 4k warmup steps. Each mini-batch consists of about 4,096 source and target tokens respec-

---

[3]The original GNMT is based on RNN, and we adapted GNMT to Transformer.
[4]See Appendix for results on RNMT, which is consistent to Transformer.

Table 2: Statistics of the training datasets for each translation tasks. These values of $D_{\mathrm{KL}}[q(z)||p(z)]$ are to some extent large, which means that MGNMT does rely on the latent variable.

| Dataset | WMT14 EN↔DE | NIST EN↔ZH | WMT16 EN↔RO | IWSLT16 EN↔DE |
|---|---|---|---|---|
| KL-annealing steps | 35k | 13.5k | 8k | 4k |
| $D_{\mathrm{KL}}[q(z)||p(z)]$ | 6.78 | 8.26 | 6.36 | 7.81 |

Table 3: BLEU scores on low-resource translation (WMT16 EN↔RO), and cross-domain translation (IWSLT EN↔DE). Note that for cross-domain translation, all models are trained with TED domain as parallel data, and NEWS domain as monolingual data if applicable, whereas these models are evaluated on the testsets of the both domains, respectively.

| Model | LOW-RESOURCE | | CROSS-DOMAIN (*para*. TED & *mono*. NEWS) | | | |
|---|---|---|---|---|---|---|
| | WMT16 EN↔RO | | TED | | NEWS | |
| | EN-RO | RO-EN | EN-DE | DE-EN | EN-DE | DE-EN |
| Transformer (Vaswani et al., 2017) | 32.1 | 33.2 | 27.5 | 32.8 | 17.1 | 19.9 |
| GNMT (Shah & Barber, 2018) | 32.4 | 33.6 | 28.0 | 33.2 | 17.4 | 20.1 |
| GNMT-M-SSL + *non-parallel* (Shah & Barber, 2018) | 34.1 | 35.3 | 28.4 | 33.7 | 22.0 | 24.9 |
| Transformer+BT + *non-parallel* (Sennrich et al., 2016b) | 33.9 | 35.0 | 27.8 | 33.3 | 20.9 | 24.3 |
| Transformer+JBT + *non-parallel* (Zhang et al., 2018) | 34.5 | 35.7 | 28.4 | 33.8 | 21.9 | 25.1 |
| Transformer+Dual + *non-parallel* (He et al., 2016a) | 34.6 | 35.7 | 28.5 | 34.0 | 21.8 | 25.3 |
| MGNMT | 32.7 | 33.9 | 28.2 | 33.6 | 17.6 | 20.2 |
| MGNMT + *non-parallel* | **34.9** | **36.1** | 28.5 | 34.2 | **22.8** | **26.1** |

tively. We trained our models on a single GTX 1080ti GPU. To avoid that the approximate posterior "collapses" to the prior that learns to ignore the latent representation while $D_{\mathrm{KL}}(q(z)||p(z))$ trends closely to zero (Bowman et al., 2016; Shah & Barber, 2018), we applied KL-annealing and word dropout (Bowman et al., 2016) to counter this effect. For all experiments, word dropout rates were set to a constant of 0.3. Honestly, annealing KL weight is somewhat tricky. Table 2 lists our best setting of KL-annealing for each task on the development sets. The translation evaluation metric is BLEU (Papineni et al., 2002). More details are included in Appendix.

## 4.1 RESULTS AND DISCUSSION

As shown in Table 3 and Table 4, MGNMT outperforms our competitive Transformer baseline (Vaswani et al., 2017), Transformer-based GNMT (Shah & Barber, 2018) and related work in both resource-poor scenarios and resource-rich scenarios.

**MGNMT makes better use of non-parallel data.** As shown in Table 3, MGNMT outperforms our competitive Transformer baseline (Vaswani et al., 2017), Transformer-based GNMT (Shah & Barber, 2018) and related work in both resource-poor scenarios.

1. *On low-resource language pairs.* The proposed MGNMT obtains a bit improvement over Transformer and GNMT on the scarce bilingual data. Large margins of improvement are obtained by exploiting non-parallel data.
2. *On cross-domain translation.* To evaluate the capability of our model in the cross-domain setting, we first trained our model on TED data from IWSLT benchmark to simulate general-domain training, and then exposed the model to NEWS non-parallel bilingual data from News Crawl to accessing target domain knowledge. As shown in Table 3, being invisible to target domain training data leads to poor performance in target domain testset (NEWS) of both Transformer and MGNMT. In this case, non-parallel data of NEWS domain contributes significantly, leading to 5.7~6.4 BLEU gains. We also conduct a case study on the cross-domain translation in Appendix.
3. *On Resource-rich scenarios.* We also conduct regular translation experiments on two resource-rich language pairs, i.e., EN↔DE and NIST EN↔ZH. As shown in Table 4, MGNMT can obtain comparable results compared to discriminative baseline RNMT and generative baseline GNMT on pure parallel setting. Our model can also achieve better performance by the aid of non-parallel bilingual data than the compared previous approaches, consistent with the experimental results in resource-poor scenarios.
4. *Comparison to other semi-supervised work.* We compare our approach with well-established approaches which are also designed for leveraging non-parallel data, including back-translation (Sennrich et al., 2016b, Transformer+BT), joint back-translation training (Zhang et al., 2018, Trans-

Table 4: BLEU scores on resource-rich language pairs.

| Model | WMT14 | | NIST | |
| --- | --- | --- | --- | --- |
| | EN-DE | DE-EN | EN-ZH | ZH-EN |
| Transformer (Vaswani et al., 2017) | 27.2 | 30.8 | 39.02 | 45.72 |
| GNMT (Shah & Barber, 2018) | 27.5 | 31.1 | 40.10 | 46.69 |
| GNMT-M-SSL + *non-parallel* (Shah & Barber, 2018) | 29.7 | 33.5 | 41.73 | 47.70 |
| Transformer+BT + *non-parallel* (Sennrich et al., 2016b) | 29.6 | 33.2 | 41.98 | 48.35 |
| Transformer+JBT + *non-parallel* (Zhang et al., 2018) | 30.0 | 33.6 | 42.43 | 48.75 |
| Transformer+Dual + *non-parallel* (He et al., 2016b) | 29.6 | 33.2 | 42.13 | 48.60 |
| MGNMT | 27.7 | 31.4 | 40.42 | 46.98 |
| MGNMT + *non-parallel* | 30.3 | 33.8 | 42.56 | 49.05 |

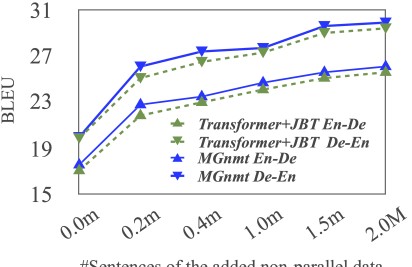

Figure 4: BLEU vs. scales of non-parallel data on IWSLT EN↔DE tasks.

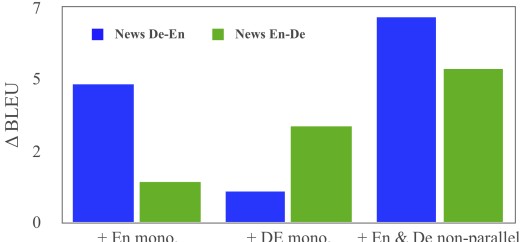

Figure 5: BLEU increments vs. adding one side monolingual (w/o interactive training) or non-parallel bilingual data for MGNMT on IWSLT EN↔DE tasks.

former+JBT), multi-lingual and semi-supervised variant of GNMT (Shah & Barber, 2018, GNMT-M-SSL), and dual learning (He et al., 2016a, Transformer+Dual). As shown in Table 3, while introducing non-parallel data to either low-resource language or cross-domain translation, all listed semi-supervised approaches gain substantial improvements. Among them, our MGNMT achieves the best BLEU score. Meanwhile, in resource-rich language pairs, the results are consistent. We suggest that because the jointly trained language model and translation model could work coordinately for decoding, MGNMT surpasses joint back-translation and dual learning. Interestingly, we can see that the GNMT-M-SLL performs poorly on NIST EN↔ZH, which means parameters-sharing is not quite suitable for distant language pair. These results indicate its promising strength of boosting low-resource translation and exploiting domain-related knowledge from non-parallel data for cross-domain scenarios.

**MGNMT is better at incorporating language model in decoding** In addition, we find from Table 5 that simple interpolation of NMT and external LM (separately trained on target-side monolingual data) (Gulcehre et al., 2015, Transformer-LM-FUSION) only produces mild effects. This can be attributed to the unrelated probabilistic modeling, which means that a more naturally integrated solution like MGNMT is necessary.

Table 5: Incorporating LM for decoding (IWSLT task).

| Model | EN-DE | DE-EN |
| --- | --- | --- |
| MGNMT: dec. w/o LM | 21.2 | 24.6 |
| MGNMT: dec. w/ LM | **22.8** | **26.1** |
| Transformer | 17.1 | 19.9 |
| Transformer+LM-FUSION | 18.4 | 21.1 |

**Comparison with noisy channel model reranking (Ng et al., 2019)** We compare MGNMT with the noisy channel model reranking (Ng et al., 2019, NCMR). NCMR uses $\log p(y|x) + \lambda_1 \log p(x|y) + \lambda_2 \log p(y)$ to rerank the translation candidates obtained from beam search, where $\lambda_1 = 1$ and $\lambda_2 = 0.3$, which are similar to our decoding setting. As shown in Table 6, NCMR is indeed effective and easy-to-use. But MGNMT still works better. Specifically, the advantage of the unified probabilistic modeling in MGNMT not only improves the effectiveness and efficiency of exploiting non-parallel data for training, but also enables the use of the highly-coupled language models and bidirectional translation models at testing time.

Table 6: Comparison with NCMR (IWSLT task).

| Model | EN-DE | DE-EN |
| --- | --- | --- |
| MGNMT + *non-parallel* | 22.8 | 26.1 |
| Transformer+BT w/ NCMR (w/o) | 21.8 (20.9) | 25.1 (24.3) |
| GNMT-M-SSL w/ NCMR (w/o) | 22.4 (22.0) | 25.6 (24.9) |

**Effects of non-parallel data.** We conduct experiments regarding the scales of non-parallel data on IWSLT EN↔DE to investigate the relationship between benefits and data scales. As shown in Figure 4, as the amount of non-parallel data increases, all models become strong gradually. MGNMT outperforms Transformer+JBT consistently in all data scales. Nevertheless, the growth rate decreases probably due to noise of the non-parallel data. We also investigate if one side of non-parallel data could benefit both translation directions of MGNMT. As shown in Figure 5, we surprisingly find that only using one side monolingual data, for example, English, could also improve English-to-German translation a little bit, which meets our expectation.

**Effects of latent variable $z$.** Empirically, Figure 6 shows gains become little when KL term gets close to 0 ($z$ becomes uninformative), while too large KL affects negatively; meanwhile, Table 2 shows that the values of $D_{\mathrm{KL}}[q(z)||p(z)]$ are relatively reasonable; besides, decoding from a zero $z$ leads to large drops. These suggest that MGNMT learns a meaningful bilingual latent variable, and heavily relies on it to model the translation task. Moreover, MGNMT adds further improvements to decoding by involving language models that condition on the meaningful semantic $z$. (Table 5). These pieces of evidence show the necessity of $z$.

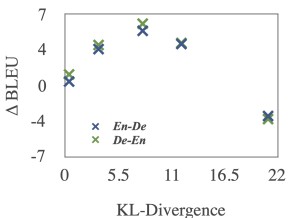

Figure 6: $\Delta$BLEU wrt $D_{\mathrm{KL}}$.

**Speed comparison** MGNMT introduces extra costs for training and decoding compared to Transformer baseline. When being trained on parallel data, MGNMT only slightly increases the training cost. However, the training cost regarding non-parallel training is larger than vanilla Transformer because of the on-fly sampling of pseudo-translation pairs, which is also the cost of joint back-translation and dual learning. As shown in Table 7, we can see that on-fly sampling implies time-consumption, MGNMT takes more

Table 7: Training (hours until early stop) and decoding cost comparison on IWSLT task. All the experiments are conducted on a single 1080ti GPU.

| Model | Training (hrs) | Decoding |
|---|---|---|
| Transformer | ∼17 | 1.0× |
| Transformer+BT | ∼25 | 1.0× |
| GNMT-M-SSL | ∼30 | 2.1× |
| Transformer+JBT | ∼34 | 1.0× |
| Transformer+Dual | ∼52 | 1.0× |
| MGNMT | ∼22 | 2.7× |
| MGNMT + *non-parallel* | ∼45 | 2.7× |

training time than joint back-translation but less than dual learning. One possible way to improve the efficiency may be to sample and save these pseudo-translation pairs in advance to the next epoch of training.

As for inference time, Transformer+{BT/JBT/Dual} are roughly the same as vanilla Transformer because essentially they do not modify decoding phase. Apart from this, we find that the decoding converges at 2∼3 iterations for MGNMT, which leads to ∼2.7× time cost as the Transformer baseline. To alleviate the sacrifice of speed will be one of our future directions.

**Robustness of noisy source sentence** We conduct experiments on noisy source sentence to investigate the robustness of our models compared with GNMT. The experimental setting is similar to Shah & Barber (2018), i.e., each word of the source sentence has a 30% chance of being missing. We conduct experiments on WMT En-De. As shown in Table 8, MGNMT is more robust than GNMT with noisy source input. This may be attributed to

Table 8: Comparison on robustness of noisy source sentence.

| Model | GNMT | MGNMT |
|---|---|---|
| En-De | 27.5 | 27.7 |
| De-En | 31.1 | 31.4 |
| En-De (noisy) | 19.4 | 20.3 |
| De-En (noisy) | 23.0 | 24.1 |

the unified probabilistic modeling of TMs and LMs in MGNMT, where the backward translation and language models are naturally and directly leveraged to better "denoise" the noisy source input. Nevertheless, the missing content in the noisy source input is still very hard to recover, leading to a large drop to all methods. Dealing with noisy input is interesting and we will leave it for future study.

## 5 CONCLUSION

In this paper, we propose the mirror-generative NMT model (MGNMT) to make better use of non-parallel data. MGNMT jointly learns bidirectional translation models as well as source and target

language models in a latent space of the shared bilingual semantics. In such a case, both translation directions of MGNMT could simultaneously benefit from non-parallel data. Besides, MGNMT can naturally take advantage of its learned target-side language model for decoding, which leads to better generation quality. Experiments show that the proposed MGNMT consistently outperforms other approaches in all investigated scenarios, and verify its advantages in both training and decoding. We will investigate whether MGNMT can be used in completely unsupervised setting in future work.

## 6 ACKNOWLEDGEMENTS

We would like to thank the anonymous reviewers for their insightful comments. Shujian Huang is the corresponding author. This work is supported by National Science Foundation of China (No. U1836221, 61672277), National Key R&D Program of China (No. 2019QY1806).

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

# A   LEARNING FROM NON-PARALLEL DATA: DERIVATION

We first take the target marginal probability $\log p(y^{(t)})$ for example to show its deviation. Inspired by Zhang et al. (2018), we introduce $y^{(t)}$'s translation $x$ in source language as intermediate hidden variable, and decompose $\log p(y^{(t)})$ as:

$$
\begin{aligned}
\log p(y^{(t)}) = \log \sum_x p(x, y^{(t)}) &= \log \sum_x Q(x) \frac{p(x, y^{(t)})}{Q(x)} \\
&\geq \sum_x Q(x) \log \frac{p(x, y^{(t)})}{Q(x)} \quad \text{(Jenson inequality)} \\
&= \sum_x Q(x) \log p(x, y^{(t)}) - Q(x) \log Q(x)
\end{aligned}
\tag{8}
$$

In order to make the equal sign to be valid in Equation (8), $Q(x)$ must be the true *tgt2src* translation probability $p^*(x|y^{(t)})$ , which can be approximated by MGNMT through $p^*(x|y^{(t)}) = p(x|y^{(t)}) = \frac{p(x, y^{(t)})}{p^*(y^{(t)})} = \frac{1}{T}\mathbb{E}_z[p(x, y^{(t)}|z)]$ via Monte Carlo sampling[5]. Analogously, the intermediate hidden variable $x$ is the translation of $y^{(t)}$ given by MGNMT itself (described in Section 3.2), which produces a pair of *pseudo* parallel sentences $\langle x, y^{(t)} \rangle$. This is similar to the back-translation (Sennrich et al., 2016b), which requires an externally separate *tgt2src* NMT model to provide the synthetic data other than the unified model itself as in MGNMT.

Remember that we have derived the low-bound of $\log p(x, y)$ in Equation (2). As a result, we now get the lower bound of $\log p(y^{(t)})$ as $\mathcal{L}(y^{(t)}; \theta_y, \theta_{xy}, \phi)$ by

$$
\begin{aligned}
\log p(y^{(t)}) \geq \mathcal{L}(y^{(t)}; \theta_y, \theta_{xy}, \phi) = \mathbb{E}_{p(x|y^{(t)})} \big[ \mathbb{E}_{q(z|x,y^{(t)};\phi)} [ \frac{1}{2} \{ \log p(x|z) + p(x|y^{(t)}, z) \\
+ \log p(y^{(t)}|z) + \log p(y^{(t)}|x, z) \} ] \\
- D_{KL}[q(z|x, y^{(t)}; \phi)||p(z)] - \log p^*(x|y^{(t)}) \big]
\end{aligned}
\tag{9}
$$

Since $p(x|y^{(t)}, z)$, $p(x|z)$ and $p^*(x|y^{(t)})$ are irrelevant to parameters $\{\theta_y, \theta_{xy}\}$, $\mathcal{L}(y^{(t)}; \theta_y, \theta_{xy}, \phi)$ could be simplified on a optimization purpose, namely:

$$
\begin{aligned}
\mathcal{L}(y^{(t)}; \theta_y, \theta_{xy}, \phi) = \mathbb{E}_{p(x|y^{(t)})} \big[ \mathbb{E}_{q(z|x,y^{(t)};\phi)} [ \frac{1}{2} \{ \log p(y^{(t)}|z; \theta_y) + \log p(y^{(t)}|x, z; \theta_{xy}) \} ] \\
- D_{KL}[q(z|x, y^{(t)}; \phi)||p(z)] \big]
\end{aligned}
\tag{10}
$$

The lower-bound $\mathcal{L}(y^{(t)}; \theta_y, \theta_{xy}, \phi)$ of $\log p(y^{(t)})$ serves as a training objective to optimize $\{\theta_y, \theta_{xy}, \phi\}$. Likewise, the lower bound of the likelihood on the target marginal probability $\log p(x^{(s)})$ could be derived as:

$$
\begin{aligned}
\mathcal{L}(x^{(s)}; \theta_x, \theta_{yx}, \phi) = \mathbb{E}_{p(y|x^{(s)})} \big[ \mathbb{E}_{q(z|x^{(s)},y;\phi)} [ \frac{1}{2} \{ \log p(x^{(s)}|z; \theta_x) + \log p(x^{(s)}|y, z; \theta_{yx}) \} ] \\
- D_{KL}[q(z|x^{(s)}, y; \phi)||p(z)] \big]
\end{aligned}
\tag{11}
$$

# B   IMPLEMENTATION DETAILS

We follow the GNMT (Shah & Barber, 2018) to implement our MGNMT. For machine translation, we have a source sentence $x = \langle x_1, \ldots, x_{L_x} \rangle$ and a target sentence $y = \langle y_1, \ldots, y_{L_y} \rangle$. As aforementioned, MGNMT consists of a variational *src2tgt* and a *tgt2src* translation models ($\text{TM}_{x \to y}(\theta_{xy})$,

---

[5]In order to make the equal sign to be valid in Equation 8, $Q(x)$ must satisfy the following condition $\frac{p(x, y^{(t)})}{Q(x)} = c$, where $c$ is a constant and does not depend on $x$. Given $\sum_x Q(x) = 1$, $Q(x)$ can be calculated as:

$$
Q(x) = \frac{p(x, y^{(t)})}{c} = \frac{p(x, y^{(t)})}{\sum_x p(x, y^{(t)})} = p^*(x|y^{(t)})
$$

where $p^*(x|y^{(t)})$ denotes the true *tgt2src* translation probability, while the target marginal probability $p^*(y) = c = \frac{1}{T}$ due to the assumption that the target sentences in $\mathcal{D}_y$ are i.i.d.

$\text{TM}_{x \to y}(\theta_{yx})$), as well as a source and a target variational language model($\text{LM}_x(\theta_x)$,$\text{LM}_y(\theta_y)$). These four components are conditioned on a shared inference model $q(z|x, y; \phi)$ as approximate posterior. The overall architecture is shown in Figure 3.

Now, we first introduce the implementations based on RNMT, which is similar to (Shah & Barber, 2018). Then we introduce the Transformer-based variant.

## B.1 RNMT-BASED MGNMT

**Language Model** Let's take the target language model $\text{LM}_y(\theta_y)$ as an example. $\text{LM}_y(\theta_y)$ models the computation of $p(y|z; \theta_y)$, which is implemented by a GRU-based (Cho et al., 2014) RNNLM (Mikolov et al., 2010) with the latent variable $z$ as additional input. The probabilities $p(y|z)$, for $i = 1, ..., L_x$ are factorized by:

$$p(y|z) = \prod_{j}^{L_y} p(y_j|y_{<t}, z) = \text{softmax}(E(y_j)^\top \mathbf{W}_y h_j^y) \tag{12}$$

where $\mathbf{W}_y$ is a learnable linear transformation matrix, $E(y_j)$ is the embedding of $y_j$, and the hidden state $h_j^y$ is computed as:

$$h_j^y = \text{GRU}(h_{t-1}^y, [z; E(y_{t-1})]) \tag{13}$$

where $[\cdot; \cdot]$ is a concatenation operation.

Likewise, the source language model $\text{LM}_x(\theta_x)$ models $p(x|z; \theta_x)$ in a mirror way.

**Translation Model** Let's take the *src2tgt* translation model $\text{TM}_{x \to y}(\theta_{xy})$ as an example. $\text{TM}_{x \to y}(\theta_{xy})$ models the computation of $p(y|x, z; \theta_{xy})$, which is implemented by the variational variant of the widely-used RNMT (Bahdanau et al., 2015). RNMT uses an encoder-decoder framework. The conditional probabilities $p(y|x, z)$, for $i = 1, ..., L_x$ are factorized by:

$$p(y|x, z) = \prod_{j}^{L_y} p(y_j|y_{<t}, x, z) = \text{softmax}(E(y_j)^\top \mathbf{U}_y s_j^y) \tag{14}$$

where $\mathbf{U}_y$ is a learnable linear transformation matrix, and the decoder hidden state $s_j^y$ is computed as:

$$\tilde{s}_j^y = \text{GRU}(s_{t-1}^y, [z; E(y_{t-1})]) \tag{15}$$

$$c_j^y = \sum_{i}^{L_x} \alpha_{ji} v_i^x, \quad \alpha_{ji} = \text{softmax}(a(\tilde{s}_j^y, v_i^x)) \tag{16}$$

$$s_j^y = \text{GRU}(\tilde{s}_j^y, c_j^y) \tag{17}$$

where $v_i^x$ is the $i$-th encoder hidden state, $c_j^y$ is the attentive context vector, which is a weighted average of the source hidden states by attentive weight $\alpha_{ji}$ given by the attention model $a$. The encoder hidden state $v_i^x$ is modeled by a bidirectional GRU (Schuster & Paliwal, 1997; Cho et al., 2014):

$$v_i^x = \overleftrightarrow{\text{BiGRU}}(v_{i\pm1}^x, E(x_{i\pm1})) \tag{18}$$

Likewise, the *tgt2src* translation model $\text{TM}_{x \to y}(\theta_{yx})$) models $p(x|z; \theta_x)$ in a mirror way.

**Inference Model** The inference model $q(z|x, y; \phi)$ serves as an approximate posterior, which is a diagonal Gaussian:

$$q(z|x, y; \phi) = \mathcal{N}(\boldsymbol{\mu}_\phi(x, y), \boldsymbol{\Sigma}_\phi(x, y)) \tag{19}$$

We first map the sentences $x$, and $y$ to a sentence representation vector using a bidirectional GRU, followed by an average pooling, respectively:

$$\bar{r}^{\mathrm{x}} = \frac{1}{L_{\mathrm{x}}} \sum_{i}^{L_{\mathrm{x}}} \overleftrightarrow{\mathrm{BiGRU}}(r_{i\pm1}^{\mathrm{x}}, E(x_{i\pm1})) \tag{20}$$

$$\bar{r}^{\mathrm{y}} = \frac{1}{L_{\mathrm{y}}} \sum_{j}^{L_{\mathrm{y}}} \overleftrightarrow{\mathrm{BiGRU}}(r_{t\pm1}^{\mathrm{y}}, E(y_{t\pm1})) \tag{21}$$

where $\bar{r}^{\mathrm{x}}$ and $\bar{r}^{\mathrm{y}}$ is the fixed-length sentence vector which is the average of the hidden states of the bidirectional GRU of $x$ and $y$, respectively. We then parameterize the inference model by:

$$q(z|x,y;\phi) = \mathcal{N}(\mathbf{W}^{\mu}[\bar{r}^{\mathrm{x}};\bar{r}^{\mathrm{y}}], \mathrm{diag}(\exp(\mathbf{W}^{\Sigma}[\bar{r}^{\mathrm{x}};\bar{r}^{\mathrm{y}}]))) \tag{22}$$

## B.2 TRANSFORMER-BASED MGNMT

Theoretically, MGNMT is independent from neural architectures we choose. As for Transformer-based MGNMT, we substitute the translation models from RNMT to Transformer (Vaswani et al., 2017), which is also extended to condition on latent semantic. The language models and inference model remain the same.

## B.3 HYPERPARAMETERS

RNMT-based MGNMT adopts 512-dimensional GRUs, 512-dimensional word embeddings, and a 100-dimensional latent variable $z$. As for Transformer-based MGNMT, we use the same configurations as `transformer-base` in Vaswani et al. (2017). The embeddings of the same language are shared in the MGNMT in our implementations. For KL-annealing (Bowman et al., 2016), we multiply the KL divergence term by a constant weight, which we linearly anneal from 0 to 1 over the initial steps of training. The KL-annealing steps are sensitive to languages and the amount of dataset. We include the KL-annealing steps of best results for each language in the paper.

## B.4 IMPLEMENTATION OF OTHER BASELINES

Back-translation (Sennrich et al., 2016b, BT), joint back-translation (Zhang et al., 2018, JBT), and dual learning (He et al., 2016a, Dual) are effective training strategies which do not depend on specific architecture. Suppose that we have monolingual data $\mathcal{D}_{\mathrm{x}}$ and $\mathcal{D}_{\mathrm{y}}$, and bilingual parallel $\mathcal{D}_{\mathrm{xy}}$. Note that the forward and backward TMs here are all Transformer or RNMT.

- **BT**: To train $\mathrm{TM}_{\mathrm{x}\to\mathrm{y}}$, we first petrain a backward translation model $\mathrm{TM}_{\mathrm{y}\to\mathrm{x}}$. And then we use $\mathrm{TM}_{\mathrm{y}\to\mathrm{x}}$ to translate $\mathcal{D}_{\mathrm{x}}$ into a pseudo source corpus $\mathcal{D}_{\mathrm{x}'}$ by beam search ($b = 2$), and $\mathcal{D}_{\mathrm{x}'}$ and $\mathcal{D}_{\mathrm{y}}$ form the pseudo parallel corpus, namely $\mathcal{D}_{\mathrm{x}'\mathrm{y}}$. We finally use the collection of $\mathcal{D}_{\mathrm{x}'\mathrm{y}}$ and $\mathcal{D}_{\mathrm{xy}}$ to train $\mathrm{TM}_{\mathrm{x}\to\mathrm{y}}$. The BT training for $\mathrm{TM}_{\mathrm{y}\to\mathrm{x}}$ is similar by alternating the language.

- **JBT**: JBT is an extension of BT in an alternative and iterative manner. 1) We first pretrain $\mathrm{TM}_{\mathrm{x}\to\mathrm{y}}$ and $\mathrm{TM}_{\mathrm{y}\to\mathrm{x}}$ on $\mathcal{D}_{\mathrm{xy}}$, respectively. 2) We use $\mathrm{TM}_{\mathrm{y}\to\mathrm{x}}/\mathrm{TM}_{\mathrm{x}\to\mathrm{y}}$ to generate pseudo parallel corpora $\mathcal{D}_{\mathrm{x}'\mathrm{y}}/\mathcal{D}_{\mathrm{xy}'}$, respectively. 3) We then re-train $\mathrm{TM}_{\mathrm{x}\to\mathrm{y}}/\mathrm{TM}_{\mathrm{y}\to\mathrm{x}}$ on the collection of $\mathcal{D}_{\mathrm{xy}}$ and $\mathcal{D}_{\mathrm{x}'\mathrm{y}}/\mathcal{D}_{\mathrm{xy}'}$ for 1 epoch, respectively. So now we have a pair of better $\mathrm{TM}_{\mathrm{x}\to\mathrm{y}}$ and $\mathrm{TM}_{\mathrm{y}\to\mathrm{x}}$. 4) We finally repeat 2) and 3) with the better TMs until training converges.

- **Dual**: 1) We first pretrain $\mathrm{TM}_{\mathrm{x}\to\mathrm{y}}$ and $\mathrm{TM}_{\mathrm{y}\to\mathrm{x}}$ on $\mathcal{D}_{\mathrm{xy}}$, respectively, and $\mathrm{LM}_{\mathrm{x}}$ and $\mathrm{LM}_{\mathrm{y}}$ on $\mathcal{D}_{\mathrm{x}}$ and $\mathcal{D}_{\mathrm{y}}$ respectively. Note that in the following training process, the LMs are fixed. 2) To train TMs from monolingual corpora, the rest of the training process follows He et al. (2016a) to iteratively and alternatively optimize the language model reward and reconstruction reward. Our implementation is heavily inspired by `https://github.com/yistLin/pytorch-dual-learning`.

## C Experiments on Rnmt

### C.1 Identical Set of Experiments as Transformer

We show experiments on Rnmt in Table 9 and 10, which shows the consistent trending as Transformer-based experiments. These results suggest that MGnmt is architecture-free, which can theoretically and practically be adapted to arbitrary sequence-to-sequence architecture.

### C.2 Comparison with Gnmt in Its Original Setting.

The lack of official Gnmt codes and their manually created datasets makes it impossible for us to directly compare MGnmt with Gnmt in their original setting. This is why we initially resorted to standard benchmark datasets. Nevertheless, we try to conduct such comparisons (Table 11). We followed Shah & Barber (2018) to conduct English-French experiments. The parallel data are provided by Multi UN corpus. Similar to Shah & Barber (2018), we created a small, medium and large amount of parallel data, corresponding to 40K, 400K and 4M sentence pairs, respectively. We created validation set of 5K and test set of 10K sentence pairs. For non-parallel data, we used the News Crawl articles from 2009 to 2012. **Note** that in Shah & Barber (2018), there is a monolingual corpora consisting 20.9M monolingual sentences used for English, which is too large and time-consuming. Here we used 4.5M monolingual sentences for English and French, respectively. As shown in Table 11, MGnmt still outperforms Gnmt.

Table 9: BLEU scores on low-resource translation (Wmt16 En↔Ro), and cross-domain translation (Iwslt En↔De).

| Model | Low-Resource | | Cross-Domain (*para*. Ted & *mono*. News) | | | |
|---|---|---|---|---|---|---|
| | Wmt16 En↔Ro | | Ted | | News | |
| | En-Ro | Ro-En | En-De | De-En | En-De | De-En |
| Rnmt | 29.3 | 29.9 | 23.1 | 28.8 | 13.7 | 16.6 |
| Gnmt | 30.0 | 30.7 | 23.4 | 29.4 | 13.8 | 16.9 |
| Gnmt-M-Ssl + *non-parallel* | 31.6 | 32.5 | 23.6 | 29.6 | 17.5 | 22.0 |
| Rnmt-BT + *non-parallel* | 31.0 | 31.7 | 23.7 | 29.9 | 16.9 | 21.5 |
| Rnmt-JBT + *non-parallel* | 31.7 | 32.3 | 24.0 | 30.1 | 17.6 | 22.1 |
| Rnmt-Dual + *non-parallel* | 31.9 | 32.5 | 23.4 | 29.6 | 17.3 | 21.9 |
| Rnmt-Lm-Fusion + *non-parallel* | 29.5 | 30.3 | - | - | 14.1 | 17.0 |
| MGnmt | 30.4 | 31.2 | 23.7 | 29.8 | 13.8 | 17.0 |
| MGnmt + *non-parallel* | **32.5** | **32.9** | 24.2 | 30.4 | **18.7** | **23.3** |

## D Case Study

As shown in Table 12, we can see that without being trained on in-domain (News) non-parallel bilingual data, the baseline Rnmt shows obvious style mismatches phenomenon. Although all the enhanced methods alleviate this domain inconsistency problem to some extent, MGnmt produces the best in-domain-related translation.

Table 10: BLEU scores on resource-rich language pairs. We report results of Newstest2014 testset for Wmt14, and MT03 testset for Nist.

| Model | Wmt14 | | Nist | |
|---|---|---|---|---|
| | En-De | De-En | En-Zh | Zh-En |
| Rnmt (Bahdanau et al., 2015) | 21.9 | 26.0 | 31.77 | 38.10 |
| Gnmt (Shah & Barber, 2018) | 22.3 | 26.5 | 32.19 | 38.45 |
| Gnmt-M-Ssl + *non-parallel* (Shah & Barber, 2018) | 24.8 | 28.4 | 32.06 | 38.56 |
| Rnmt-BT + *non-parallel* (Sennrich et al., 2016b) | 23.6 | 27.9 | 32.98 | 39.21 |
| Rnmt-JBT + *non-parallel* (Zhang et al., 2018) | 25.2 | 28.8 | 33.60 | 39.72 |
| MGnmt | 22.7 | 27.9 | 32.61 | 38.88 |
| MGnmt + *non-parallel* | **25.7** | **29.4** | **34.07** | **40.25** |

Table 11: BLEU scores of RNMT-based experiments on English-French using similar settings as Shah & Barber (2018). Numbers in parentheses are quoted from GNMT paper. **Note** that because we used 4.5M English monolingual sentences instead of the original 20.9M (too time-consuming), the reproduced results of "GNMT-M-SSL" are a bit lower.

| Model | 40K | | 400K | | 4M | | avg. | Δ |
|---|---|---|---|---|---|---|---|---|
| | EN-FR | FR-EN | EN-FR | FR-EN | EN-FR | FR-EN | | |
| RNMT | 11.86 | 12.30 | 27.81 | 28.13 | 37.20 | 38.00 | 25.88 | 0 |
| GNMT | 12.32(12.47) | 13.65(13.84) | 28.69(28.98) | 29.51(29.41) | 37.64(37.97) | 38.49(38.44) | 26.72 | 0.84 |
| GNMT-M-SSL + *non-parallel* | 18.60 (20.88) | 18.92 (20.99) | 35.90(37.37) | 36.72(39.66) | 38.75(39.41) | 39.30(40.69) | 31.37 | 5.49 |
| MGNMT | 12.52 | 14.02 | 29.38 | 30.10 | 38.21 | 38.89 | 27.19 | 1.31 |
| MGNMT + *non-parallel* | **19.25** | **19.33** | **36.73** | **37.57** | **39.45** | **39.98** | **32.05** | **6.17** |

Table 12: An example from IWSLT DE-EN cross-domain translation. In this case, all the models were first trained on parallel bilingual data from TED talks (IWSLT2016), and exposed to non-parallel bilingual data of NEWS domain (News Crawl).

| | |
|---|---|
| Source | "Die Benzinsteuer ist einfach nicht zukunftsfähig", so Lee Munnich, ein Experte für Verkehrsgesetzgebung an der Universität von Minnesota. |
| Reference | "The *gas tax* is *just* not sustainable", said Lee Munnich, *a transportation policy expert* at the University of Minnesota. |
| RNMT | "The **gasoline** tax is **simply** not sustainable," so Lee Munnich, an expert on the University of Minnesota. |
| RNMT-BT | "The gas tax is **simply** not sustainable," **so** Lee Munnich, an **expert on traffic legislation** at the University of Minnesota . |
| RNMT-JBT | "The gas tax is just not sustainable," say Lee Munnich, an **expert on traffic legislation** at the University of Minnesota . |
| MGNMT | "The *gas tax* is *just* not sustainable," said Lee Munnich, *an traffic legislation expert* at the University of Minnesota. |

