# OpenReview forum: "Mirror-Generative Neural Machine Translation"
_ICLR.cc/2020/Conference — Accept (Talk)_

### Official Review · AnonReviewer2 · 2019-10-23
**Official Blind Review #2**

**Rating:** 8

**Review:**

This paper proposes an approach to neural MT in which the joint (source, target) distribution is modeled as an average over two different factorizations: target given source and source given target. This gives rise to four distributions - two language models and two translation models - which are parameterized separately but conditioned on a common variational latent variable.  The model is trained on parallel data using a standard VAE approach. It can additionally be trained on non-parallel data in an approach similar to iterated back-translation at sentence-level granularity, but with language and translation model probabilities for observed sentences coupled by the latent variable. Inference iterates between sampling a latent variable given the current best hypothesis, and using beam search plus rescoring to find a new best hypothesis given the current latent variable. The approach is evaluated in several different scenarios (low- and high-resource, domain adaptation - trained on parallel data only or parallel plus monolingual data) and found to generally outperform previous work on generative NMT and iterated back-translation.

Strengths: clearly written, well motivated, very comprehensive experiments comparing to relevant baselines.

Weaknesses: somewhat incremental relative to Shah and Barber (Neurips 2018), results are only marginally positive, framework is probably too cumbersome to justify widespread adoption based on the results.

I think the paper should be accepted. Although it’s not highly original, it ties together three strands of work in a principled way: joint models, variational approaches, and back-translation / dual learning. The increment over Shah and Barber is bolstered by the addition of back-translation, which gives substantial improvements when using non-parallel data; and to a lesser extent by the argument about the advantage of separate models for distant language pairs. Using all possible LMs and TMs coupled with a latent variable feels like an area that was inevitably going to get explored, and this paper does a good job of it. Although the gains over the baselines are not overly compelling, they are quite systematic, indicating that the advantage is probably real, albeit slight. The authors are also to be commended on their use of not just the Shah and Barber baseline, but also the back-translation-based techniques, which are generally stronger competitors when monolingual data is incorporated.

Further comments/questions:

Why are there no results for Transformer+Dual in table 4? This omission looks odd, since Transformer+Dual was the strongest baseline in table 3.

Please add implementation details for the Transformer+{BT,JBT,Dual} baselines.

It was surprising not to see robustness experiments like Shah and Barber’s dropped source words, since robustness to source noise could be one of the advantages of having an explicit model of the source sentence.

A few additional suggestions for related work: noisy channel approaches (eg, The Neural Noisy Channel, Yu et al, ICLR 2017); decipherment (eg, Beyond parallel data: Joint word alignment and decipherment improves machine translation, EMNLP 2014 - yes, from SMT days, but still); other joint modeling work (KERMIT: Generative Insertion-Based Modeling for Sequences, Chan et al, 2019).

Consider dropping the “1+1 > 2” metaphor. It’s not clear to me exactly what it means, or what it adds to the paper.

“deviation” is used in a couple places where you probably meant “derivation”?

Line 6 in algorithm 2 should use both forward and backward scores for rescoring.


**Experience Assessment:**

I have published in this field for several years.

**Review Assessment: Checking Correctness Of Derivations And Theory:**

I assessed the sensibility of the derivations and theory.

**Review Assessment: Checking Correctness Of Experiments:**

I assessed the sensibility of the experiments.

**Review Assessment: Thoroughness In Paper Reading:**

I read the paper at least twice and used my best judgement in assessing the paper.

---

> ### Author Response · Authors · 2019-11-15
> **Response to Reviewer 2**
>
> Thanks very much for your valuable comments!
>
> Q1: About missing Transformer+Dual in table 4
> A1: Thanks for your kind notes. We have RNMT+Dual in similar experiments in the Appendix showing that MGNMT (RNMT-based) outperforms RNMT+Dual in WMT En-De and NIST Zh-En. We provide results of Transformer+Dual here, which has also included in the updated paper draft:
>
>  ----------------------------------------------------------------------------------
>  Model                                            EN-DE  DE-EN  EN-ZH  ZH-EN
>  ----------------------------------------------------------------------------------
>  Transformer+Dual +NP             29.6      33.2      42.13    48.60
>  MGNMT (Transformer) + NP    30.3      33.8      42.56    49.05
>  ----------------------------------------------------------------------------------
> * NP: non-parallel
>
> As shown in the results, MGNMT is consistently better than dual learning on these high resource tasks.
>
> Q2: Experiments on noisy source sentence
> A2: Thank you so much for your nice suggestion! We conduct experiments on noisy source sentence to investigate the robustness of our models compared with GNMT. The experimental setting is similar to Shah & Barber's: each word of the source sentence has a 30% chance of being missing.
>
> Given the noisy source sentence, MGNMT first samples a draft target translation as usual. And then,
> 1) we compute the latent variable z given the current (noisy) source and target sentences;
> 2) remember that MGNMT is symmetric, we can easily use TM(Y->X|Z) and LM(X|Z) to find a corrected source sentence (run line 5, Alg. 2).
> 3) given the better-corrected source sentence, we can find a better target translation (run line 4-6, Alg.2).
> 4) repeat 1) - 3) until converge.
>
> We conduct experiments on WMT En-De. The results are as follow:
>
> ------------------------------------------------------------------------------------------------
> Model                                EN-DE   DE-EN   EN-DE (noisy)   DE-EN (noisy)
> ------------------------------------------------------------------------------------------------
> GNMT (Transformer)        27.5      31.1                19.4           23.0
> MGNMT (Transformer)    27.7      31.4                20.3            24.1
> ------------------------------------------------------------------------------------------------
>
> As shown in the above table, our model is more robust than GNMT with noisy source input. This may be attributed to the unified probabilistic modeling of TMs and LMs in MGNMT, where the backward translation and language models are naturally and directly leveraged to better "denoise" the noisy source input. Nevertheless, the missing content in the noisy source input is still very hard to recover, leading to a large drop to all methods. Dealing with noisy input is interesting and we will leave it for future study.
>
> Q3: About missing references
> A3: Thanks. We will add content to discuss these related work in the next version.
>
> Q4: About line 6 in Alg 2
> A4: Yes, it should be both forward and backward scores and we actually implemented this exactly as you suggested. We will fix it in the next version.
>
> Q5: About implementation details for the Transformer+{BT,JBT,Dual}
> A5: We have added the implementation details for them in the revised draft.

---

### Official Review · AnonReviewer1 · 2019-10-24
**Official Blind Review #1**

**Rating:** 8

**Review:**

This work proposes a new translation model that combines translation models in two directions and language modelss in two languages by sharing a latent semantic representation. The basic idea to joint modeling of translations conditioning on the latent representations and the parameters are learned by generating pseudo translations in two directions. Decoding is also carefully designed by interchanging sampling in two directions in a greedy fashion. Empirical results show consistent gains when compared with heuristic methods to generate pseudo data, e.g., back translation.

It is an interesting work on proposing a unified framework to translation by conditioning on a shared latent space in four models. It is not only rivaling heuristic methods to generate pseudo data, but surpassing competitive Transformer baselines.

Other comment:

- It is a bit confusing that MGNMT was not experimented with Transformer, though the paper and appendix describe that it is easy to use the Transformer in the MGNMT setting.

**Experience Assessment:**

I have published in this field for several years.

**Review Assessment: Checking Correctness Of Derivations And Theory:**

I assessed the sensibility of the derivations and theory.

**Review Assessment: Checking Correctness Of Experiments:**

I assessed the sensibility of the experiments.

**Review Assessment: Thoroughness In Paper Reading:**

I read the paper at least twice and used my best judgement in assessing the paper.

---

> ### Author Response · Authors · 2019-11-15
> **Response to Reviewer 1**
>
> Thanks very much for your comments!
>
> Q1: About experimenting MGNMT with Transformer
> A1: All experimental results in the experiment section are implemented based on Transformer. We also give results implemented on RNMT, which are listed in the Appendix. We will make it clearer in the revision. Sorry for the confusion.

---

### Official Review · AnonReviewer3 · 2019-10-25
**Official Blind Review #3**

**Rating:** 8

**Review:**

In this paper, the authors propose MGNMT (Mirror Generative NMT) which aims to integrate s2t, t2s, source and target language models in a single framework. They lay out the details of their framework and motivate the need for leveraging monolingual data in both source and target directions. They also talk about related work in this space. Finally, they perform experiments on low and high resource tasks. They also investigate certain specific phenomena like effect of non-parallel data, effect of target LM during decoding, and effect of adding one side monolingual data.

Pros:
- Overall, the paper was clearly written and well motivated. The authors clearly lay out their new framework and establish it for the reader.
- The set of experiments are very detailed and the authors make sure to compare against all semi-supervised works like BT, JBT and Dual learning.
- The set of analyses at the end was also interesting and tried to dig deeper in certain phenomena.
- All training details and hyperparameters have been laid it in the paper.

Cons:
- For all the additional complexity, this newly proposed method only slightly outperforms other semi-supervised methods like BT, JBT & Dual learning as seen in Tables 3 and 4.
- The authors could have been more upfront about training and inference costs of their proposed framework and compared it to the other setups. For example, decoding costs 2.7x more than a vanilla transformer. A comparison of decoding and training costs of all methods would have provided the right balance between complexity and quality. This additional complexity might outweigh the gains obtained in some cases.

Rating Justification:
Despite the con of added complexity, I like the formulation of the new joint framework and I think this will serve as a good starting point for others to push in this direction further. Hence, I want to see this paper accepted.

Minor comments:
last para of section 1: first line is too big. Please break into multiple lines.
"Exploiting non-parallel data for NMT" - second para, please cite Dong et. al and Johnson et. al who also share al parameters and vocab in a single model.
Page 5, section 3.2, second para - line 1 please rephrase.

**Experience Assessment:**

I have published in this field for several years.

**Review Assessment: Checking Correctness Of Derivations And Theory:**

I assessed the sensibility of the derivations and theory.

**Review Assessment: Checking Correctness Of Experiments:**

I carefully checked the experiments.

**Review Assessment: Thoroughness In Paper Reading:**

I read the paper at least twice and used my best judgement in assessing the paper.

---

> ### Author Response · Authors · 2019-11-15
> **Response to Reviewer 3**
>
> Thanks very much for your insightful comments!
>
> Q1. About training and decoding costs of MGNMT
> A1: Yes, MGNMT introduces extra costs for training and decoding compared to Transformer baseline.  When being trained on parallel data, our model only slightly increases the training cost. However, the training cost regarding non-parallel training is larger than vanilla Transformer because of the on-fly sampling of pseudo-translation pairs, which is also the cost for JBT and dual learning. We list the training time of each model on IWSLT task as follow, which has also been added in the revised paper draft:
>
> --------------------------------------------------------------------------------------------------------------------
>       Model              training time to converge (hrs, on a single 1080ti)       decoding
> --------------------------------------------------------------------------------------------------------------------
> Transformer                                  ~17h                                                                1x
> Transformer+BT +NP                   ~25h                                                                1x
> Transformer+JBT +NP                  ~34h                                                                1x
> Transformer+Dual +NP               ~52h                                                                1x
> GNMT-M-SSL +NP                         ~30h                                                                2.1x
> MGNMT (Transformer)                ~22h                                                                2.7x
> MGNMT (Transformer) + NP       ~45h                                                                2.7x
> --------------------------------------------------------------------------------------------------------------------
> *NP: non-parallel
>
> We can see that on-fly sampling leads to time-consumption, and MGNMT takes more training time than JBT but less than Dual. One possible way to improve the efficiency may be to sample and save these pseudo-translation pairs in advance to the next iteration of training.
>
> As for inference time, Transformer+{BT/JBT/Dual} are roughly the same as vanilla Transformer because essentially they are different strategies for training Transformer which do not modify the decoding phase. Meanwhile, MGNMT requires ~2.7x time for decoding because MGNMT needs iterative decoding for several iterations.
>
> Q2: Typos and related works
> A2: Thanks. We will fix typos and missing citations in the next version.

---

### Comment · Area_Chair1 · 2019-10-31
**Relationship to other methods considering bidirectional translation scores at test time?**

I've taken a look at this paper, and the method looks interesting! However, I had a question related to previous work. Because one of the fundamental ideas of the paper is that it should consider translation in both directions, including at test time, the most relevant baselines seem to be ones that do so.

For example, in the paper Tu et al. (2017) and Cheng et al. (2017) are cited as being essentially the same as the proposed methods. How would the proposed method compare to these theoretically and empirically?

In addition, there are much simpler methods of simply training bidirectional translation systems and combining them together at test time using reranking, or other more complicated search algorithms (Yu et al. 2016, Hoang et al. 2017). In particular, reranking with bidirectional scores is already being used, for example in FAIR's high-scoring submission for the WMT shared task this year, so it is obviously a practical and widely-understood method (Ng et al. 2019).

Could the authors please clarify, at the very least, why the proposed method would theoretically be superior to these other methods? It would also be ideal if it was possible to empirically compare with another simple baseline, e.g. generating a large n-best list with a forward GNMT model, and re-ranking with the combination of forward and backward GNMT scores like was done in (Ng et al. 2019).

* Yu, Lei, et al. "The neural noisy channel." arXiv preprint arXiv:1611.02554 (2016).
* Hoang, Cong Duy Vu, Gholamreza Haffari, and Trevor Cohn. "Towards decoding as continuous optimisation in neural machine translation." Proceedings of the 2017 Conference on Empirical Methods in Natural Language Processing. 2017.
* Nathan Ng, Kyra Yee, Alexei Baevski, Myle Ott, Michael Auli and Sergey Edunov. "Facebook FAIR’s WMT19 News Translation Task Submission." Proceedings of WMT 2019. 2019.

---

> ### Author Response · Authors · 2019-11-15
> **Response to Area Chair 1**
>
> Thank you very much for your comments!
>
> Q1: Comparison with Cheng et al. (2016) and Tu et al. (2017)
> * Correction: Sorry we incorrectly cited the Cheng et al. (2017) in our paper, which is supposed to be Cheng et al (2016) instead.
>
> Cheng et al (2016) exploit parallel corpus, and source and target monolingual corpora to jointly learn forward and backward TMs, which is equivalent to dual learning with a subtle difference (no LM reward). Given the empirical comparison that our method outperforms dual learning, our method should also be more effective than Cheng et al.
>
> Differently, Tu et al. (2017) propose to use reconstruction regularization to improve forward TM, where they do not have a backward TM. They also show that reconstruction regularization is helpful to rerank the translation candidates. Our proposed MGNMT jointly models 4 different models, i.e., bidirectional TMs and LMs, which is very different from Tu et al. and can naturally take advantage of reranking by reconstructive score.
>
>
> Q2. Comparison with other methods incorporating backward TM at test time (Yu et al. 2016a, Hoang et al. 2017, and Ng et al. 2019)
> A2: Thanks. We will add comparisons with them in the revised draft.
>
> The similarity with these works: leveraging bidirectional TMs to improve decoding at testing time.
>         - For Yu et al. (2016a), they propose a Neural Noisy Channel (NNC) model that leverages forward TM (direct model, $p(y|x)$), backward TM (channel model, $p(x|y)$, relying on a more sophisticated Segment-to-Segment Transduction model (Yu et al. 2016b)) and language model ($p(y)$) to improve the faithfulness of the output to the input, which also requires a more complicated Viterbi-based search algorithm for decoding.
>         - As for Hoang et al. (2017), they propose to treat decoding as a continuous optimization problem and use gradient-based methods (Exponentiated Gradient or SGD) to maximize arbitrary local/global objectives, e.g., a combination of forward and backward TMs scores.
>         - Simpler than the above both works, Ng et al. (2019) propose Noisy Channel Model Reranking, where an independently trained backward TM ($p(x|y)$) to rerank the N-best list of the translations of the forward TM, i.e., the reranking score function is: $p_(y|x) + \lambda_1 * p(x|y) + \lambda_2 * p(y)$.
>
> Differences with these related work: the unified probabilistic modeling of MGNMT, which benefits both training and testing.
>         - That is to say, we have four unified and coupled models, which are jointly learned with the same latent space. As a result, this could make all these models cooperate well for decoding at testing time. Another advantage is that we can achieve iterative boosting, where the reranking in Ng et al. (2019) can be seen as a part of one iteration of our proposed decoding method. In our iterative decoding framework, the translations reranked by backward TM at the current iteration can help forward TM to generate better translations at the next iterations.
>
>
> Q3: Empirical comparisons with simple baselines  as suggested
> A3: As you suggested, we compare MGNMT with the Noisy Channel Model Reranking (NCMR) method in Ng et al. (2019) where, the reranking score function become: $p_(y|x) + \lambda_1 * p(x|y) + \lambda_2 * p(y)$, where $\lambda_1=1$ and $\lambda_2 = 0.3$ (similar to our decoding setting). We conduct comparisons on IWSLT task. The results are shown as follow, which have also been added to the revised draft:
>
> ------------------------------------------------------------------------------------------
> Model                                                          EN-DE             DE-EN
> ------------------------------------------------------------------------------------------
> Transformer+ BT w/ NCMR (w/o)          21.8 (20.9)    25.1 (24.3)
> GNMT-M-SSL w/ NCMR (w/o)                 22.4 (22.0)    25.6 (24.9)
> MGNMT (Transformer)                            22.8               26.1
> ------------------------------------------------------------------------------------------
>
> We find that the reranking method in Ng et al. (2019) is indeed effective and easy-to-use. But MGNMT still works better. This can be because that the advantage of the unified probabilistic modeling in MGNMT not only improves the effectiveness and efficiency of exploiting non-parallel data for training, but also enables the use of the highly-coupled language models and bidirectional translation models at testing time. The idea of MGNMT may also be useful to improve the multi-lingual translation.
>
> * Cheng, et al. "Semi-Supervised Learning for Neural Machine Translation." ACL, 2016.
> * Cheng, et al. "Joint Training for Pivot-based Neural Machine Translation." IJCAI, 2017.
> * Yu, et al. "The neural noisy channel." arXiv preprint arXiv:1611.02554. 2016a.
> * Yu, et al. "Online segment to segment neural transduction". EMNLP. 2016b.

---

> > ### Comment · Area_Chair1 · 2019-11-15
> > **Thank you for the clarification!**
> >
> > These are very nice updates, thank you. I feel the paper will be significantly stronger and more complete after it is modified to reflect these additional theoretical and empirical comparisons.

---

> > > ### Author Response · Authors · 2019-11-15
> > > **Thank you for your important suggestions!**
> > >
> > > Your comments are really helpful. Thank you again.

---

### Public Comment · ~MarcAurelio_Ranzato1 · 2020-05-02
**comparison to related work**

Dear Authors,
 could you please elaborate more on the relation of your work to these two prior works:

[1] G. Lample, M. Ott, A. Conneau, L. Denoyer, M. Ranzato " Phrase-Based & Neural Unsupervised Machine Translation". EMNLP 2018

and

[2] P.J. Chen, J. Shen, M. Le, V. Chaudhary, A. El-Kishky, G. Wenzek, M. Ott, M. Ranzato " Facebook AI's WAT19 Myanmar-English Translation Task Submission ".  Workshop on Asian Translation at EMNLP 2019

In particular, [1] trained jointly using p(y|x), p(x|y) and p(y) and p(x) (these last two terms in the form of a denoising auto-encoder). This seems similar to your method except for the particular choice of models and the variational formulation (which is anyway quite tricky to get to work in these applications as you stated).

[2] uses both parallel and monolingual data and decodes off-line with noisy channel decoding (so in this case the language model is only used at decoding time).

Your only comment to an earlier version of [1] was " However, they may still fail to apply to distant language pairs" but [1] was shown to work to some extent to distant languages like En-Ur.

Thanks!

---

> ### Author Response · Authors · 2020-05-14
> **Response to related work**
>
> Thanks for your kind reminder of the related work. We've updated our paper to include the discussion of the two papers.
>
> As you mentioned, there are connections between [1] and our model. The most important one is that [1] and MGNMT jointly model both translation models and language models. However, there are still several differences:
>
> 1. To model bilingual shared latent representations between source and target languages,  [1] chooses an implicit manner by sharing encoder features to act like an "interlingual", while MGNMT adopts an explicit manner by modeling the latent space under variational formulation.
> 2. MGNMT can be trained in an end-to-end manner, while [1] follows the pipeline of three principles (initialization-language modeling-back translation) and requires incorporating an extra PBSMT.
>
> As for distant language pairs, our main concern is that sharing vocabularies (BPEs) may make a little effect, which is one of the essential parts of [1]. We show experiments on Chinese-English in Table 4 of our paper, where GNMT-M-SSL (sharing BPE vocabulary) underperformed a simple back-translation method. This was not the case for related languages like English-German, where sharing vocabulary indeed contributed greatly to related languages. Hence, these results made us concerned about the potential issue of non-overlapping alphabets for distant languages.
>
>
> As for [2], we compared with another similar paper [3] also using noisy channel model reranking from Facebook as well, which was shown to work very well. Please refer to the results in Table 6, where we find that MGNMT still works better.
>
> __
>
> [3]  Nathan Ng, Kyra Yee, Alexei Baevski, Myle Ott, Michael Auli, and Sergey Edunov. Facebook fairs wmt19 news translation task submission. In WMT, 2019.

---

### Decision · Program_Chairs · 2019-12-19

**Decision:**

Accept (Talk)

**Comment:**

This paper proposes a novel method for considering translations in both directions within the framework of generative neural machine translation, significantly improving accuracy.

All three reviewers appreciated the paper, although they noted that the gains were somewhat small for the increased complexity of the model. Nonetheless, the baselines presented are already quite competitive, so improvements on these datasets are likely to never be extremely large.

Overall, I found this to be a quite nice paper, and strongly recommend acceptance, perhaps as an oral presentation.